# Restore Text First, Enhance Image Later: Two-Stage Scene Text Image Super-Resolution with Glyph Structure Guidance

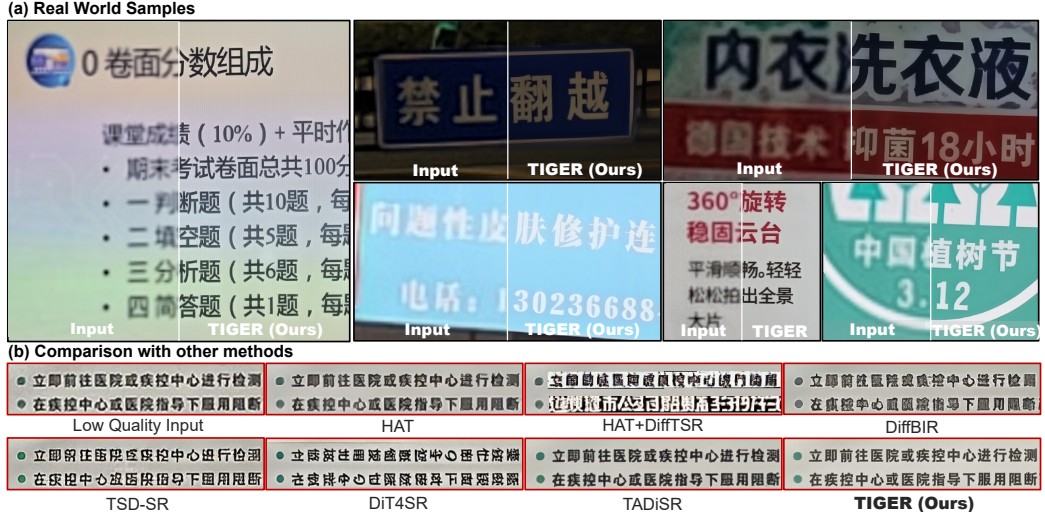

Figure 1: We present **TIGER** (**T**ext–**I**mage **G**uided sup**E**r-**R**esolution), a novel framework for scene text super-resolution. Its 'text-first, image-later' paradigm ensures accurate glyph restoration and consistently high overall image fidelity and visual quality.

## Abstract

Current generative super-resolution methods show strong performance on natural images but distort text, creating a fundamental trade-off between image quality and textual readability. To address this, we introduce **TIGER** (**T**ext–**I**mage **G**uided sup**E**r-**R**esolution), a novel two-stage framework that breaks this trade-off through a *"text-first, image-later"* paradigm. TIGER explicitly decouples glyph restoration from image enhancement: it first reconstructs precise text structures and then uses them to guide subsequent full-image super-resolution. This glyph-to-image guidance ensures both high fidelity and visual consistency. To support comprehensive training and evaluation, we also contribute the **UltraZoom-ST** (UltraZoom-Scene Text), the first scene text dataset with extreme zoom (×**14.29**). Extensive experiments show that TIGER achieves **state-of-the-art** performance, enhancing readability while preserving overall image quality.

## 1 Introduction

Scene Text Image Super-Resolution is a critical problem in computer vision for its vital role in car navigation (Li & Cui, 2025), scene understanding (Kil et al., 2023; Deshmukh et al., 2024), and document enhancement (Souibgui et al., 2023). It seeks to restore a high-quality super-resolution (SR) image from a degraded low-resolution (LR) input while preserving the correct text meaning. Unlike general image super-resolution for natural scenes, where plausible texture synthesis or detail hallucination is acceptable and does not alter the semantic meaning of the image, visual texts exhibit extremely low tolerance to structural errors. This is especially evident in Chinese, where a minor stroke distortion or omission may entirely alter the meaning (Yu et al., 2023; Li et al., 2023a).

Table 1: Dataset comparison. 'Non-Latin characters' indicates inclusion of languages such as Chinese, 'Multi-lines' denotes images with multiple text lines, 'Real-world degradation ($> \times 4$)' refers to captured LR with zoom beyond $\times 4$. TextZoom (Wang et al., 2020) lacks non-Latin text, CTR (Yu et al., 2021) lacks multi-line text, and Real-CE (Ma et al., 2023) has only mild degradation. UltraZoom-ST covers all, providing a more challenging benchmark.

| Content | TextZoom | CTR | Real-CE | UltraZoom-ST (Ours) |
|---|---|---|---|---|
| Non-Latin characters | ✗ | ✓ | ✓ | ✓ |
| Multi-lines | ✗ | ✗ | ✓ | ✓ |
| Real-World Degradation ($> \times 4$) | ✗ | ✗ | ✗ | ✓ |

In recent years, with the rapid development of generative models such as diffusion (Ho et al., 2020; Dhariwal & Nichol, 2021; Saharia et al., 2022; Ramesh et al., 2022; Rombach et al., 2022; Chang et al., 2023; Zhang et al., 2024b), image super-resolution has increasingly relied on their strong generative priors to recover missing details from low-resolution (LR) inputs (Wu et al., 2024b; Yu et al., 2024; Lin et al., 2024; Wu et al., 2024a; Dong et al., 2025a; Duan et al., 2025; Hu et al., 2025). Although these approaches can effectively restore fine-grained natural textures (e.g., grass, leaves), they often distort text regions, turning them into gibberish as shown in Fig.1 (b). Some researchers attempt to address this by processing only text regions (Zhang et al., 2024a; Li et al., 2023a). While these methods improve readability notably, the absence of global background constraints often introduces style inconsistencies and block artifacts between text and background. The unsatisfactory performance of the present methods can be attributed to several reasons. Firstly, there is a deficiency in scene text paired data. Current scene text super-resolution datasets (Wang et al., 2020; Yu et al., 2021; Ma et al., 2023) are limited in degradation and focused on textline annotations, making it difficult for models to learn the mapping from low- to high-resolution text. Moreover, current methods primarily enhance overall image quality but fail to capture fine-grained glyph structures. This limitation is amplified for Chinese characters, due to their complex glyph designs and relatively low saliency in images. Consequently, models tend to under-represent and collapse glyphs into oversimplified, averaged forms, producing overlapping or distorted characters. Ultimately, current approaches face a persistent trade-off between maintaining readability and ensuring high image quality. *Our key observation is that these goals need not be antagonistic if text and non-text are explicitly handled differently. Text structures can be reconstructed with dedicated mechanisms and then used to guide full-image restoration, ensuring coherent style without artifacts.*

Building on this insight, we introduce **TIGER** (**T**ext–**I**mage **G**uided sup**E**r-**R**esolution), a progressive two-stage paradigm for scene text super-resolution built on the principle of *"restoring text structure first, enhancing the whole image later."* Unlike methods that rely on a single generative prior, TIGER explicitly separates the treatment of text and non-text regions: a diffusion-based local text refiner focuses on reconstructing fine-grained stroke geometry in text regions, ensuring glyph fidelity and structural consistency. The recovered text structures are then injected as conditional guidance into the subsequent full-image restoration stage, steering global super-resolution to harmonize text and background while suppressing artifacts and preserving overall visual quality.

Moreover, to address the issue of data scarcity and enable comprehensive evaluation (Wang et al., 2020; Ma et al., 2023), we introduce the **UltraZoom-ST** (UltraZoom-Sence Text) benchmark dataset, the first scene text dataset that contains extreme zooming mode ($\times 14.29$), providing extra challenging scenarios for the field. Content differences are outlined in Table 1. It includes high-quality 5,036 LR–HR pairs captured at multiple focal lengths (ranging from 14mm to 200mm), with 49,675 text lines in total. Each pair comes with detailed annotations, including detection boxes and text transcripts, to support both training and evaluation. The dataset includes diverse scenarios such as shop signs, posters, and documents. It also covers varying lighting conditions, including daylight, indoor lighting, and nighttime, offering a challenging yet practical benchmark for the field. To ensure reliable alignment across extreme focal lengths, we adopt a coarse-to-fine cascade alignment strategy: coarse alignment is performed via global geometric transformation, and residual misalignments are corrected through local refinement. This process achieves accurate LR–HR alignment, thereby ensuring both the quality of the dataset and the reliability of our experiments. The main contributions are summarized as follows:

1. We propose TIGER, the first two-stage scene text super-resolution framework that introduces a novel *'text-first, image-later'* paradigm to decouple glyph restoration from image enhancement, improving both readability and visual quality.

2. We introduce UltraZoom-ST, the first scene text benchmark with extreme zooming ($\times 14.29$), offering well-aligned, richly annotated LR–HR pairs for comprehensive evaluation under challenging and diverse real-world conditions.

3. Extensive experiments on both Real-CE and UltraZoom-ST show the proposed method outperforms prior state-of-the-art models, particularly in preserving text structure fidelity.

## 2 RELATED WORKS

**Real-World Image Super-Resolution.** Real-world image super-resolution (Real-SR) aims to reconstruct high-resolution (HR) images from low-resolution (LR) inputs degraded under uncontrolled real-world conditions. Early GAN-based methods, such as BSRGAN (Zhang et al., 2021) and Real-ESRGAN (Wang et al., 2021), synthesize degradations through random combinations of known distortions and use adversarial training for restoration. While they generate natural-looking images, their instability and insensitivity to fine details limit their ability to recover structural elements, especially text. Recent work introduces diffusion models into Real-SR (Yue et al., 2023; Wang et al., 2024b), improving perceptual quality but still struggling with complex structures like scene text. StableSR (Wang et al., 2024a) and DiffBIR (Lin et al., 2024) apply ControlNet (Zhang et al., 2023) to condition generation on LR inputs, while PASD (Yang et al., 2024) and SeeSR (Wu et al., 2024b) incorporate high-level semantics to enhance fidelity. SUPIR (Yu et al., 2024) scales training with large image-text pairs and introduces degradation-robust encoders. Other approaches, including OSEDiff (Wu et al., 2024a) and TSD-SR (Dong et al., 2025b), directly apply the diffusion process on LR images and distill models for one-step sampling. DiT-SR (Cheng et al., 2025), Dream-Clear (Ai et al., 2024), and DiT4SR (Duan et al., 2025) adopt diffusion transformers (DiT) for Real-SR. Despite enhancing perceptual fidelity, these methods often overlook accurate text structures. TADiSR (Hu et al., 2025) addresses Chinese scene text super-resolution by using Kolors as the base model and aggregating cross-attention maps for text structure supervision. However, constrained by the resolution of cross-attention, it struggles to restore small or severely degraded text.

**Text Image Super-Resolution.** Text image super-resolution (Text-SR) restores textual content from cropped images containing isolated words or text lines. Early methods (Dong et al., 2015) apply general SR architectures such as SRCNN (Dong et al., 2014) to enhance OCR performance on LR. TextSR (Wang et al., 2019) introduces GANs with text recognition loss, while PlugNet (Mou et al., 2020) and STT (Chen et al., 2021) jointly train SR and recognition modules for more discriminative features. TSRN (Wang et al., 2020) introduces the TextZoom dataset and incorporates an edge-aware module to preserve text details, while TATT (Ma et al., 2022) uses a global attention module to handle irregular text layouts. Recent methods leverage stronger generative priors. MAR-CONet (Li et al., 2023a) employs StyleGAN priors and a glyph structure codebook for realistic text reconstruction. DiffTSR (Zhang et al., 2024a) employs latent diffusion to separately denoise text and text-image components. Despite progress in text structure restoration, existing methods lack global background constraints, causing style inconsistencies and block artifacts between text and background.

## 3 METHODOLOGY

### 3.1 ARCHITECTURE OVERVIEW

Current image super-resolution methods focus on enhancing overall image quality but often fail to accurately preserve glyph structures, leading to distorted text in super-resolved images, as shown in Fig. 1 (b). Conversely, text image super-resolution methods improve text readability but do not retain the global semantic information of the background, causing incoherence between text and non-text regions. To leverage the strengths of both approaches, we introduce the TIGER framework. As illustrated in Fig. 2, the TIGER framework is composed of 2 stages, the **Text Restoration stage** and the **Text Restoration stage**. In the text restoration stage, we extract the text regions from the LR input $x_L \in \mathbb{R}^{H \times W \times C}$ and feed them into the glyph structure restoration model to restore the text structure based on the text region of the LR input and the predicted text. We then reassemble the text structures to their original positions to obtain a text mask $\hat{x}_m \in \mathbb{R}^{H \times W \times C}$. In the image enhancement stage, the text mask and LR input are then processed by a ControlNet-like network to obtain the enhanced SR output $\hat{x}_H \in \mathbb{R}^{H \times W \times C}$. The following sections detail each stage.

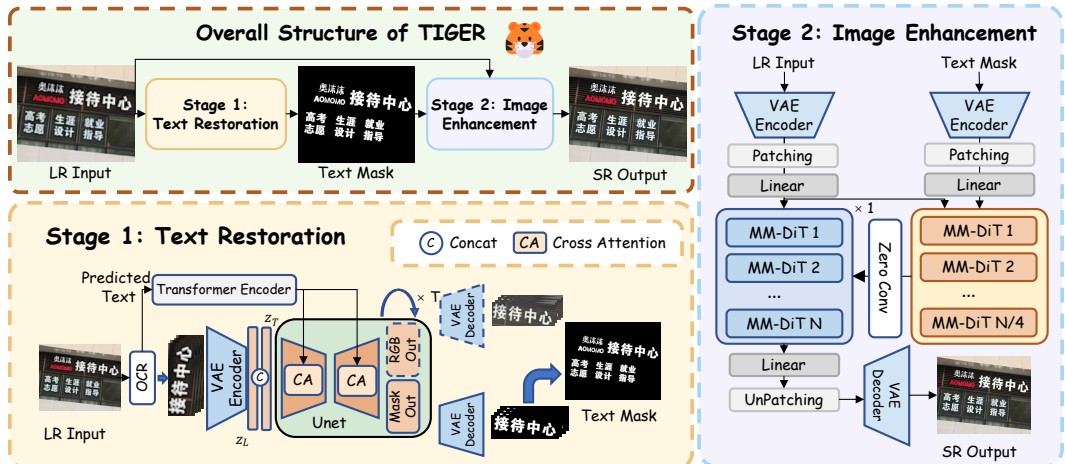

Figure 2: The framework of TIGER, which includes the Text Restoration stage (stage 1) and the Image Enhancement stage (stage 2). Stage 1 refines text regions to recover accurate glyph structures. Stage 2 uses these structures to guide full-image restoration for coherent text and background.

## 3.2 TEXT RESTORATION STAGE

**Text Restoration Pipeline.** Existing methods like Hi-SAM (Ye et al., 2024) work well on clean high-resolution text but fail on the incomplete, distorted text in low-resolution images. To address this, we propose a region-level text restoration pipeline shown in Fig. 2 built on Rombach et al. (2022) to recover glyph structures $x_m \in \mathbb{R}^{H \times W \times C}$. An OCR detector first localizes text regions $\{x_L^0, ..., x_L^{N-1}\}$ and extracts their contents $\{y^0, ..., y^{N-1}\}$ as semantic conditions. Each region $\tilde{x}_L$ is encoded by VAE (Kingma & Welling, 2013) into $\tilde{z}_L \in \mathbb{R}^{h \times w \times c}$, concatenated with noise $z_T \in \mathbb{R}^{h \times w \times 2c}$, and iteratively denoised by a UNet $\epsilon_\theta$ into two branches: $z_{t-1}^{RGB} \in \mathbb{R}^{h \times w \times c}$ for appearance and $z_{t-1}^m \in \mathbb{R}^{h \times w \times c}$ for structure, which are merged as $z_{t-1}$. The text content $y$ is embedded into $c_{te}$ and fused into $z_t$ via cross-attention to guide structure recovery. After $T$ denoising steps, the mask branch output $z_0^m$ is decoded into $\tilde{x}_m$, and all restored regions are assembled into the final text mask $\hat{x}_m$. Focusing on real text reconstruction enables the model to capture glyph structures without non-text interference and reduces sensitivity to text saliency.

**Training Strategy.** The scarcity of real-world segmentation masks for degraded text forces reliance on synthetic data, yet its artificial degradation limits generalization. Annotating real degraded samples, however, is labor-intensive and costly. To marry synthetic precision with real-world degradation, we propose a two-phase training strategy. In Phase 1, both synthetic and real data are included in training. This allows the model to generate text masks from the LR image and capture real-world degradation patterns. However, noisy masks in real data degrade mask quality. To address this, we freeze the RGB out block and the mask out block and train the UNet with only synthetic data to refine the quality of output text masks in Phase 2. The training objective is described as follows:

$$\mathcal{L} = \lambda_{td}\mathcal{L}_{td} + \lambda_{Seg}\mathcal{L}_{Seg}, \tag{1}$$

where $\mathcal{L}_{td}$, $\mathcal{L}_{Seg}$, $\lambda_{td}$, and $\lambda_{Seg}$ denote the text-control diffusion loss, segmentation-oriented loss, and their corresponding hyperparameters. The text-control diffusion loss is formulated as follows:

$$\mathcal{L}_\epsilon = \mathbb{E}_{z_0, \tilde{z}_L, c_{te}, t, \epsilon \sim \mathcal{N}(0,1)} \left[ \|\epsilon - \epsilon_\theta(z_t, \tilde{z}_L, c_{te}, t)\|_2^2 \right]. \tag{2}$$

To improve the quality of the text mask, we propose a segmentation-oriented loss. Let $\varepsilon_t$ denote the noise predicted by the denoiser network $\epsilon_\theta$. Following Ho et al. (2020), $z_0^m$ can be estimated by combining the time step $t$ with the noisy latent image $z_t^m$. This estimate is subsequently passed through the VAE decoder to obtain an approximate reconstruction of the original input text mask, denoted as $x_0'^m$. In this way, text mask generation can be supervised at the pixel level. We combine Mean Squared Error (MSE), Focal, and Dice Losses, commonly used in segmentation tasks (Ye et al., 2024), to compare $x_0'^m$ with the original image $x_0^m$, as shown in the following equation:

$$\mathcal{L}_{Seg} = \|x_0'^m - x_0^m\|_2^2 + \lambda_{Focal}\text{FocalLoss}(x_0'^m, x_0^m) + \lambda_{Dice}\text{DiceLoss}(x_0'^m, x_0^m), \tag{3}$$

where $\lambda_{Focal}$ and $\lambda_{Dice}$ are their respective balancing coefficients. This combination forms a novel text-first paradigm that equips the model with both structural accuracy and real-world generalization, laying the foundation for the subsequent full-image super-resolution stage.

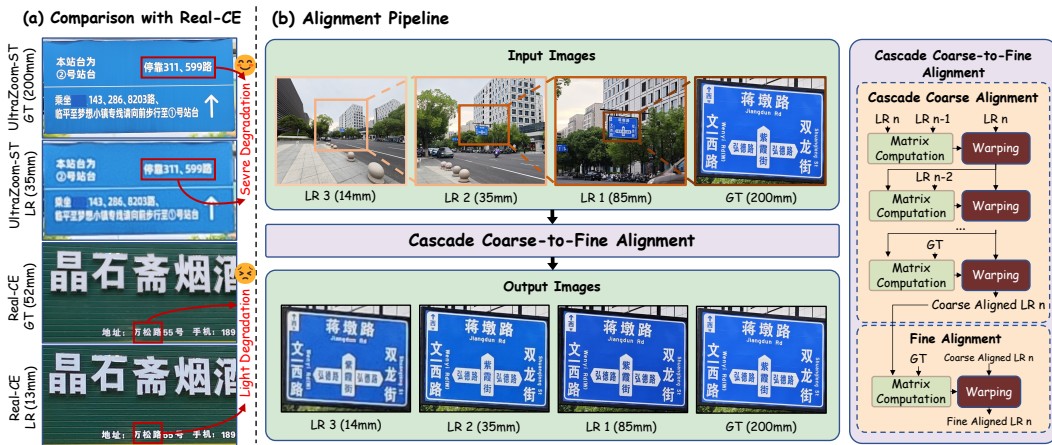

Figure 3: Overview of UltraZoom-ST. (a) Real-CE LRs show only mild degradation (red box), while UltraZoom-ST LRs exhibit stronger degradation (red box), enabling a more comprehensive evaluation. (b) Coarse-to-fine alignment: images are sorted by focal length, each warped to the next higher-focal neighbor using an estimated homography matrix, then refined to the 200 mm GT.

### 3.3 IMAGE ENHANCEMENT STAGE

**Image Enhancement pipeline.** To effectively take advantage of the generated glyph structure $\hat{x}_m$ and enhance the quality of LR $x_L$, we adopt a ControlNet (Zhang et al., 2023) network $\epsilon_\phi$. As illustrated in Fig. 2, after getting their latent representation $\hat{z}_m$ and $z_L$, we use the network to denoise the $z_L$ at $t$ timestep, using the null-text embedding $c_{Null}$. The output super-resolved latent representation $\hat{z}_H$ will be denoted as:

$$\hat{z}_H = z_L - \sigma_t \epsilon_\phi(z_L, \hat{z}_m, t, c_{Null}), \tag{4}$$

where $\sigma_t$ is a scalar determined by the predefined diffusion time step. Together with the text restoration stage, this image enhancement pipeline forms a progressive 'text-first, image after' paradigm; our design injects structure-aware control into the generative process, allowing the network to enhance global image quality without eroding the recovered text structures.

**Training Strategy.** For the quality of image super-resolution, we constrain the reconstruction loss between the predicted high-resolution image $\hat{x}_H$ decoded from $\hat{z}_H$ by $\varepsilon$ and the ground truth high-resolution image $x_H$ using a weighted sum of MSE and LPIPS losses:

$$\mathcal{L}_{img} = \lambda_{l2} \|x_H - \hat{x}_H\|_2^2 + \lambda_{LPIPS} \text{LPIPS}(x_H, \hat{x}_H), \tag{5}$$

where $\lambda_{l2}$ and $\lambda_{LPIPS}$ are balancing coefficients for different loss terms. To enhance glyph control and emphasize the glyph structure, we extract the text boundaries using Sobel operators (Roberts & Mullis, 1987). This edge loss is expressed as:

$$\mathcal{L}_{edge} = \|\text{Sobel}(x_H) - \text{Sobel}(\hat{x}_H)\|_2^2. \tag{6}$$

We combine the two loss terms and use $\lambda_{edge}$ as the balancing coefficient in the complete loss function for stage 2:

$$\mathcal{L} = \mathcal{L}_{img} + \lambda_{edge}\mathcal{L}_{edge}. \tag{7}$$

The combination of glyph-aware ControlNet conditioning and edge-constrained training objectives enables our model to preserve stroke integrity while harmonizing text and background appearance.

## 4 DATASET AND BENCHMARK

Existing datasets like Ma et al. (2023) offer only subtle degradation, as shown in Fig. 3 (a), making them insufficient for evaluating model robustness. To address the lack of both challenging and publicly available datasets tailored for scene text image super-resolution, especially for Chinese text, we introduce **UltraZoom-ST**, a challenging real-world benchmark. It is collected using ViVO X200 Ultra equipped with four fixed focal lengths (14 mm, 35 mm, 85 mm, 200 mm), enabling image

Table 2: Evaluation results on image quality. Numbers in **bold** indicate the best performance, and underscored numbers indicate the second best. TIGER (Ours) performs best on image quality.

| Methods | Real-CE | | | | | UltraZoom-ST | | | | |
|---|---|---|---|---|---|---|---|---|---|---|
| | PSNR↑ | SSIM↑ | LPIPS↓ | DISTS↓ | FID↓ | PSNR↑ | SSIM↑ | LPIPS↓ | DISTS↓ | FID↓ |
| Real-ESRGAN | 22.30 | 0.787 | 0.239 | 0.188 | 53.60 | 23.99 | 0.790 | 0.248 | 0.194 | 30.60 |
| HAT | 23.61 | 0.830 | 0.214 | 0.176 | 51.16 | 25.17 | 0.815 | 0.249 | 0.198 | 30.12 |
| MARCONet | 21.89 | 0.785 | 0.238 | 0.150 | 52.97 | 22.13 | 0.768 | 0.306 | 0.205 | 34.28 |
| SeeSR | 23.59 | 0.822 | 0.195 | 0.169 | 43.75 | 23.64 | 0.788 | 0.219 | 0.184 | 26.73 |
| SupIR | 21.78 | 0.723 | 0.310 | 0.198 | 44.94 | 23.62 | 0.754 | 0.308 | 0.207 | 27.91 |
| DiffTSR | 22.10 | 0.768 | 0.278 | 0.168 | 44.91 | 22.41 | 0.767 | 0.300 | 0.189 | 31.25 |
| DiffBIR | 22.44 | 0.747 | 0.260 | 0.201 | 46.44 | 23.67 | 0.724 | 0.262 | 0.197 | 23.10 |
| OSEDiff | 21.86 | 0.771 | 0.197 | 0.127 | 41.00 | 25.07 | 0.819 | 0.201 | 0.169 | 20.53 |
| DreamClear | 22.47 | 0.772 | 0.216 | 0.157 | 38.97 | 24.10 | 0.773 | 0.238 | 0.191 | 21.75 |
| TSD-SR | 21.43 | 0.754 | 0.220 | 0.175 | 47.21 | 22.79 | 0.757 | 0.207 | 0.194 | 24.08 |
| DiT4SR | 20.54 | 0.764 | 0.268 | 0.186 | 49.79 | 23.16 | 0.767 | 0.215 | 0.159 | 20.58 |
| TADiSR | 23.83 | 0.790 | 0.286 | 0.154 | 44.42 | 24.61 | 0.796 | 0.203 | 0.160 | 36.61 |
| **TIGER (Ours)** | **24.12** | **0.839** | **0.164** | **0.125** | **38.72** | **25.48** | **0.830** | **0.196** | **0.156** | **20.01** |

Table 3: Evaluation of image quality and text accuracy. Metrics with $cr$ are computed only on text regions. TIGER (ours) achieves the best performance in both.

| Methods | Real-CE | | | | | UltraZoom-ST | | | | |
|---|---|---|---|---|---|---|---|---|---|---|
| | $PSNR_{cr}$ | $SSIM_{cr}$ | $LPIPS_{cr}$ | $DISTS_{cr}$ | OCR-A | $PSNR_{cr}$ | $SSIM_{cr}$ | $LPIPS_{cr}$ | $DISTS_{cr}$ | OCR-A |
| Real-ESRGAN | 21.71 | 0.824 | 0.257 | 0.224 | 0.560 | 21.25 | 0.786 | 0.302 | 0.266 | 0.371 |
| HAT | 23.03 | 0.862 | 0.249 | 0.239 | 0.566 | 22.18 | 0.813 | 0.291 | 0.276 | 0.379 |
| MARCONet | 20.68 | 0.786 | 0.267 | 0.223 | 0.550 | 17.48 | 0.690 | 0.529 | 0.366 | 0.334 |
| SeeSR | 22.74 | 0.844 | 0.237 | 0.212 | 0.374 | 20.45 | 0.767 | 0.297 | 0.241 | 0.191 |
| SupIR | 19.96 | 0.778 | 0.326 | 0.263 | 0.278 | 20.10 | 0.756 | 0.344 | 0.287 | 0.236 |
| DiffTSR | 20.46 | 0.818 | 0.275 | 0.239 | 0.441 | 17.00 | 0.665 | 0.452 | 0.313 | 0.317 |
| DiffBIR | 21.20 | 0.792 | 0.278 | 0.224 | 0.376 | 20.70 | 0.757 | 0.302 | 0.245 | 0.266 |
| OSEDiff | 19.38 | 0.768 | 0.269 | 0.208 | 0.268 | 21.43 | 0.798 | 0.276 | 0.263 | 0.300 |
| DreamClear | 22.74 | 0.845 | 0.190 | 0.168 | 0.502 | 20.50 | 0.768 | 0.317 | 0.276 | 0.214 |
| TSD-SR | 19.33 | 0.763 | 0.279 | 0.234 | 0.349 | 19.42 | 0.748 | 0.296 | 0.249 | 0.266 |
| DiT4SR | 17.95 | 0.738 | 0.317 | 0.244 | 0.292 | 19.73 | 0.760 | 0.273 | 0.216 | 0.237 |
| TADiSR | 23.39 | 0.855 | 0.253 | 0.258 | 0.647 | 21.59 | 0.799 | 0.360 | 0.336 | 0.384 |
| **TIGER (Ours)** | **23.43** | **0.864** | **0.173** | **0.167** | **0.673** | **22.22** | **0.814** | **0.228** | **0.212** | **0.430** |

pairs with extreme ×14.29 zoom. It includes diverse scenes—street views, book covers, advertisements, menus, and posters—captured under varied lighting, with all text lines manually annotated. However, such extreme zoom introduces severe misalignment that breaks pixel-wise optimization methods like Cai et al. (2019), originally designed for moderate ×2 to ×4 zoom. To overcome this, we design a **Cascade Coarse-to-fine Alignment** pipeline shown in Fig. 3 (b): images are first sorted by focal length, and each low-quality image is sequentially aligned to its next higher-focal neighbor using either optimization-based registration or feature-based extractors (e.g., Lowe (1999); Künzel et al. (2025)), then refined against the 200 mm ground truth for precise alignment.

Through meticulous annotating and aligning, we obtained a total of 5,036 image pairs, with 49,675 lines of text. We set images under 200mm focal lengths as GT, and obtain 1,439, 1,798, and 1,799 pairs for ×14.29, ×5.71, and ×2.35 zooming modes, respectively. Among them, we randomly select 470, 589, and 581 pairs for evaluation under each zooming mode. Each image pair contains one or more text lines. These evaluation sets enable us to evaluate the performance of models under more complex and challenging scenarios.

For the UltraZoom-ST-benchmark, we utilize 5 evaluation metrics to examine the quality of image super-resolution on the scale of the full image. Firstly, we adopt Peak Signal-to-Noise Ratio (PSNR), Structural Similarity Index Measure (SSIM) (Wang et al., 2019), Learned Perceptual Image Patch Similarity (LPIPS) (Zhang et al., 2018), Deep Image Structure and Texture Similarity (DISTS) (Ding et al., 2020), and Fréchet Inception Distance (FID) (Heusel et al., 2017) for evaluation of image quality. Specifically, PSNR and SSIM are computed in the pixel space to quantify low-level reconstruction fidelity, LPIPS and DISTS are computed in the feature space to assess perceptual similarity, and FID is employed to evaluate the distributional discrepancy between generated and real images. However, these metrics do not directly reflect the quality or accuracy of

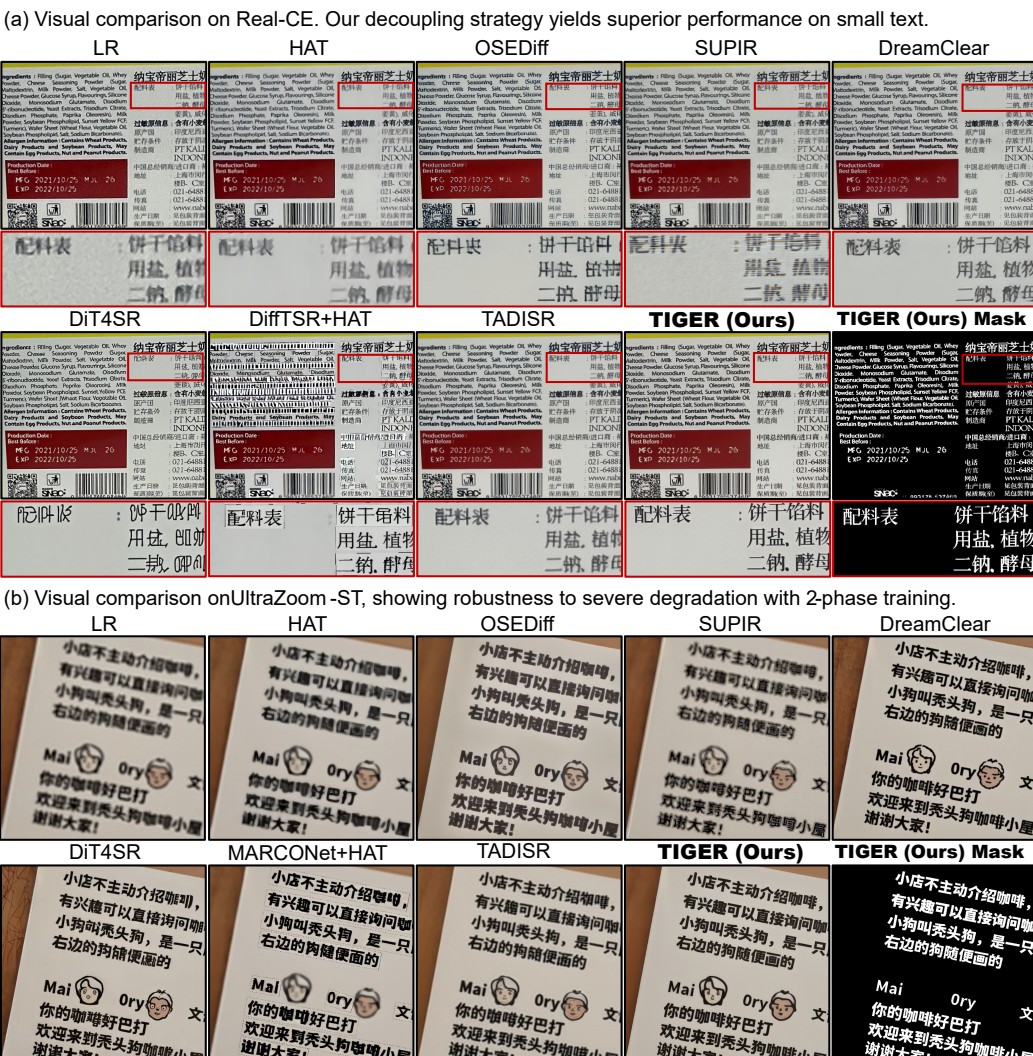

Figure 4: Qualitative Evaluation on Real-CE and UltraZoom-ST.

super-resolved text. To address this, we crop annotated text regions from the images and compute PSNR, SSIM, LPIPS, and DISTS on these cropped regions, denoted as $\text{PSNR}cr$, $\text{SSIM}cr$, $\text{LPIPS}cr$, and $\text{DISTS}cr$. For text accuracy, we apply an OCR model to recognize the text and compare its outputs with the ground-truth annotations using the Levenshtein ratio (Yujian & Bo, 2007):

$$\text{OCR-A} = (\text{Len}(s_{pred}) + \text{Len}(s_{gt}) - \text{Dist}(s_{pred}, s_{gt}))/(\text{Len}(s_{pred}) + \text{Len}(s_{gt})), \qquad (8)$$

where $s_{pred}$ stands for the predicted text sequence, $s_{gt}$ denotes the annotated ground-truth text sequence, and $\text{Dist}(\cdot, \cdot)$ is the Levenshtein distance between the text sequences.

## 5 EXPERIMENT

### 5.1 IMPLEMENTATION DETAILS

We combine synthetic data (built upon LSDIR (Li et al., 2023b) with text rendered via LBTS (Tang et al., 2023) and Real-ESRGAN (Wang et al., 2021) degradation) and real paired data from Real-CE (Ma et al., 2023) and UltraZoom-ST. Misaligned pairs in Real-CE are filtered and reannotated, yielding 337 training and 188 testing pairs. Stage 1 uses cropped text regions and is trained using an IDM-based architecture (Zhang et al., 2024a) with combined segmentation and reconstruction losses. Stage 2 is based on Stable Diffusion 3.5 (Esser et al., 2024) and employs a tile-based inference strategy (Yu et al., 2024; Hu et al., 2025); it is pretrained on synthetic data and fine-tuned on real data. More detailed information can be found in the Appendix A.3.

Table 4: Validation on UltraZoom-ST (UZST). Finetuning improves OCR-A, proving effectiveness.

| Methods | PSNR↑ | SSIM↑ | LPIPS↓ | DISTS↓ | FID↓ | OCR-A↑ |
|---|---|---|---|---|---|---|
| OSEDiff w/o UZST | 23.50 | 0.791 | **0.197** | **0.162** | 24.30 | 0.228 |
| OSEDiff w/  UZST | **25.07** | **0.819** | 0.201 | 0.169 | **20.53** | **0.300** |
| DiT4SR w/o UZST | 22.58 | 0.754 | 0.252 | 0.196 | 26.93 | 0.193 |
| DiT4SR w/  UZST | **23.17** | **0.767** | **0.215** | **0.159** | **20.58** | **0.237** |
| Ours w/o UZST | 22.96 | 0.782 | 0.341 | 0.253 | 33.78 | 0.400 |
| Ours w/  UZST | **25.48** | **0.830** | **0.196** | **0.156** | **20.01** | **0.430** |

## 5.2 COMPARISON RESULTS

### 5.2.1 QUANTITATIVE RESULTS

We evaluate existing competing methods, including GAN-based image super-resolution approaches such as Real-ESRGAN (Wang et al., 2021) and HAT (Chen et al., 2023); diffusion-based approaches such as SeeSR (Wu et al., 2024b), SupIR (Yu et al., 2024), DiffBIR (Lin et al., 2024), OSEDiff (Wu et al., 2024a), DreamClear (Ai et al., 2024), TSD-SR (Dong et al., 2025b), and DiT4SR (Duan et al., 2025); as well as text-focused reconstruction methods such as MARCONet (Li et al., 2023a), DiffTSR (Zhang et al., 2024a), and TADiSR (Hu et al., 2025). Evaluations are performed on the Real-CE Benchmark (Ma et al., 2023) and the benchmark described in Sec. 4. To ensure fairness, we fine-tune the released pre-trained models on the training sets of both benchmarks using the official code when available. Following Hu et al. (2025), we integrate the outputs of MARCONet and DiffTSR with HAT-generated results to simulate real-world application scenarios and enable comprehensive full-image evaluation. We evaluate Real-CE at the hardest difficulty level ($\times 4$), while UltraZoom-ST is evaluated across all difficulty levels—with average results reported here. Detailed per-level evaluations appear in the Appendix A.4.

As shown in Table 2 and Table 3 , TIGER outperforms competing methods on both the Real-CE and UltraZoom-ST benchmarks in terms of image quality and text accuracy. TADiSR, limited by the resolution of its cross-attention mechanism, performs poorly under severe degradation and with small text. DiffTSR and MARCONet fail to effectively handle text backgrounds and struggle with text regions that have a large width-to-height aspect ratio. In contrast, our method restores images with high quality and fine-grained text, owing to the decoupling strategy. It achieves OCR-A scores above 0.67 and 0.43, and demonstrates overall superiority in both pixel-level and perceptual accuracy.

### 5.2.2 QUALITATIVE RESULTS

Fig. 4 shows visual comparisons on Real-CE and UltraZoom-ST. Real-world degradations cause noise, blurred strokes, and distortions in the input images. GAN-based methods reduce noise but fail to fix text structure, while diffusion-based SR methods, though powerful, lack text-structure guidance and often distort strokes, sometimes making text unreadable.

Methods tailored for text image super resolution (e.g., MARCONet and DiffTSR) exhibit limitations in processing long text sequences, often generating distorted or semantically meaningless results. They also lack global semantic guidance, making it hard to blend text regions with backgrounds. As seen in Fig. 4 (a), DiffTSR+HAT (red box) produces text colors inconsistent with the original LR and SR output of HAT. In Fig. 4 (b), it even distorts the face near the word "Ory". TADiSR works well on lightly degraded images but fails under severe degradation and performs poorly on small text due to cross-attention mask resolution limits. Our method overcomes these issues, restoring high-quality glyph structures even in challenging cases.

## 5.3 ABLATION STUDY

**Effectiveness of the training set.** Table 4 reports consistent OCR-A improvements across different architectures, including the UNet-based OSEDiff and the DiT-based DiT4SR. This demonstrates that even models without explicit mechanisms for text structure modeling can still learn the mapping from low- to high-resolution text using our dataset, further validating its effectiveness. Due to their lack of a dedicated mechanism for text structure modeling, they cannot outperform our methods.

Table 5: Ablation study on Real-CE with stage 2 fixed as the baseline. From top to bottom, we compare the performance of using text masks rendered with a **standard font**, extracted using **SAM-TS**, reconstructed with **latent diffusion model** conditioned on LR, and reconstructed with our **text restoration pipeline**. Our pipeline faithfully restores glyph structures, yielding the highest accuracy.

| Text Mask Settings | PSNR↑ | SSIM↑ | LPIPS↓ | DISTS↓ | FID↓ | OCR-A↑ |
|---|---|---|---|---|---|---|
| Std Font Guidance | 23.31 | 0.786 | 0.249 | 0.164 | 41.83 | 0.553 |
| SAM-TS Extraction | 23.91 | 0.835 | 0.212 | 0.147 | 41.22 | 0.579 |
| LDM Guidance | 22.42 | 0.798 | 0.222 | 0.147 | 43.08 | 0.601 |
| **TIGER (Ours)** | **24.12** | **0.839** | **0.164** | **0.125** | **38.72** | **0.673** |

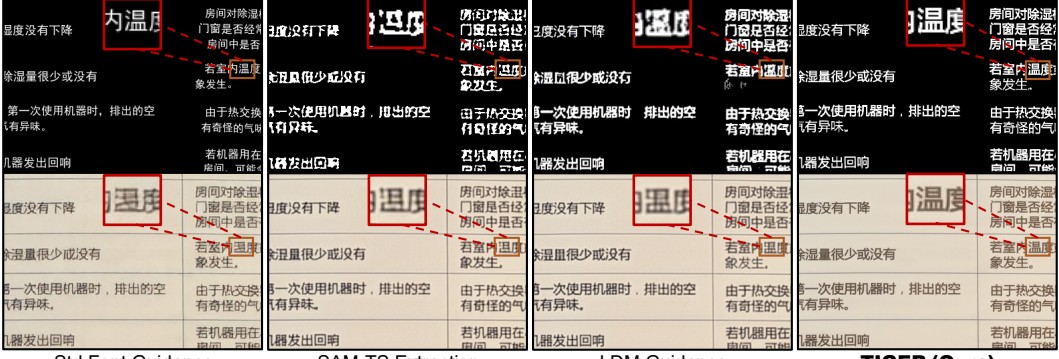

Figure 5: Qualitative Results of Ablation Study with stage 2 fixed as the baseline.

**Effectiveness of the TIGER component.** We set stage 2 as the fixed baseline for image restoration and validate the effectiveness of our framework through 3 ablated variants:

- **Standard Font Guidance.** As shown in Fig. 5 (first column), the standard font, while structurally correct, provides weak guidance due to style and position mismatches. As a result, the restored text exhibits poor accuracy compared to TIGER, with the OCR-A dropping from 0.671 to 0.553, as shown in Table 5.

- **SAM-TS Extraction Hu et al. (2025); Ye et al. (2024).** While this improves visual quality by better aligning output masks with the LR image, SAM-TS can only extract distorted structures and can't recover degraded text (Fig. 5, second column), leading to suboptimal performance compared with TIGER, with OCR-A dropping from 0.671 to 0.579.

- **LDM Guidance.** This variant uses a latent diffusion model conditioned on LR to restore text structure, achieving better accuracy by reconstructing or compensating for lost text. However, without separate learning from synthetic and real-world data, it struggles with high-quality text structures in real-world scenarios. The masks remain noisy under real-world degradations (Fig. 5, third column), leading to poorer accuracy compared to TIGER, with OCR-A dropping from 0.671 to 0.601 as shown in Table 5.

Our method leverages RGB output to jointly learn from real-world degradations and accurate synthetic masks, producing high-quality masks and achieving the highest accuracy overall.

## 6 CONCLUSION

In this paper, we delve into the extensively researched problem of scene text super-resolution. To address this challenge, we propose a novel approach called TIGER, which decouples the glyph structure restoration and image enhancement. For the restoration of text structure, we propose the text restoration pipeline, which enables a 2-phase training strategy to fully take advantage of both synthetic and real-world data. For the image enhancement, we propose the image enhancement pipeline, which effectively utilizes the glyph structure to restore both text fidelity and image details coherently in full-image super-resolution. In terms of training data and benchmark, we present the UltraLens-ST, the first real-world scene text dataset that contains extreme zooming mode, offering more challenging scenarios for the field. Extensive experiments on Real-CE and UltraLens-ST demonstrate the superiority of TIGER over existing methods.

## REPRODUCIBILITY STATEMENT

To ensure reproducibility, we have made the following efforts: (1) We will release our code and dataset. (2) We provide implementation details in Sec. 5.1 and Appendix A.3, including the training process and selection of hyper-parameters. (3) We provide details on evaluation metrics and dataset preparation in Sec. 4 and Appendix A.2, and the code and data will be made available along with it.

## ETHICS STATEMENT

This work focuses on improving scene text image super-resolution to support beneficial applications such as enhancing accessibility, document restoration, and navigation assistance. However, we acknowledge potential risks, including misuse in privacy-sensitive contexts (e.g., recovering text from personal or social media images) or unintended deployment in surveillance. To mitigate such risks, users are encouraged to combine our methods with privacy-preserving techniques such as watermarking or selective inpainting. Our UltraZoom-ST dataset was collected from public, non-sensitive scenes under ethical guidelines, without including personally identifiable or private information. We believe the societal benefits of improved text image restoration outweigh potential risks, provided that the technology is applied responsibly.

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

# A APPENDIX

## A.1 LLM ACKNOWLEDGMENTS

We thank ChatGPT (GPT-5) and DeepSeek for the help in refining the language and improving clarity. All wording and factual content were reviewed and approved by the authors.

Table 6: Evaluation Results on Both Image Quality and Text Accuracy on UltraZoom-ST.

| | Methods | PSNR↑ | SSIM↑ | LPIPS↓ | DISTS↓ | FID↓ | PSNR$_{cr}$↑ | SSIM$_{cr}$↑ | LPIPS$_{cr}$↓ | DISTS$_{cr}$↓ | OCR-A↑ |
|---|---|---|---|---|---|---|---|---|---|---|---|
| | Real-ESRGAN | 25.84 | 0.854 | 0.139 | 0.130 | 28.25 | 23.59 | 0.843 | 0.170 | 0.168 | 0.632 |
| | HAT | **27.33** | **0.873** | 0.128 | 0.133 | 27.21 | 24.76 | **0.870** | 0.139 | 0.146 | 0.643 |
| | MACRONet | 22.90 | 0.804 | 0.234 | 0.173 | 40.19 | 17.92 | 0.693 | 0.515 | 0.360 | 0.542 |
| | SeeSR | 25.02 | 0.826 | 0.167 | 0.159 | 32.31 | 21.62 | 0.796 | 0.245 | 0.207 | 0.374 |
| 85mm (×2.35) | SupIR | 25.58 | 0.823 | 0.202 | 0.160 | 29.33 | 21.91 | 0.808 | 0.239 | 0.214 | 0.351 |
| | DiffTSR | 23.48 | 0.805 | 0.230 | 0.157 | 35.77 | 17.92 | 0.688 | 0.392 | 0.285 | 0.482 |
| | DiffBIR | 25.25 | 0.793 | 0.175 | 0.152 | 24.57 | 22.26 | 0.803 | 0.218 | 0.190 | 0.391 |
| | OSEDiff | 26.38 | 0.852 | 0.153 | 0.137 | 23.92 | 22.84 | 0.833 | 0.200 | 0.204 | 0.433 |
| | DreamClear | 25.31 | 0.811 | 0.169 | 0.151 | 24.19 | 21.44 | 0.790 | 0.253 | 0.221 | 0.291 |
| | TSD-SR | 23.99 | 0.800 | 0.161 | 0.165 | 30.40 | 20.62 | 0.786 | 0.232 | 0.218 | 0.401 |
| | DiT4SR | 24.71 | 0.810 | 0.173 | 0.143 | 29.15 | 21.25 | 0.800 | 0.221 | 0.193 | 0.482 |
| | TADiSR | 26.31 | 0.858 | 0.177 | 0.149 | 30.47 | 23.55 | 0.856 | 0.175 | 0.178 | 0.581 |
| | Ours | 27.00 | 0.871 | **0.123** | **0.120** | **22.34** | 24.31 | 0.860 | **0.135** | **0.142** | **0.658** |
| | Real-ESRGAN | 23.70 | 0.785 | 0.218 | 0.184 | 42.62 | 20.86 | 0.782 | 0.272 | 0.235 | 0.348 |
| | HAT | 25.05 | 0.819 | 0.207 | 0.182 | 41.32 | 21.98 | 0.816 | 0.250 | 0.241 | 0.354 |
| | MACRONet | 22.28 | 0.774 | 0.284 | 0.199 | 49.14 | 17.43 | 0.694 | 0.523 | 0.361 | 0.316 |
| | SeeSR | 23.76 | 0.795 | 0.198 | 0.175 | 37.92 | 20.55 | 0.778 | 0.271 | 0.222 | 0.317 |
| 35mm (×5.71) | SupIR | 23.49 | 0.746 | 0.302 | 0.204 | 41.80 | 20.00 | 0.757 | 0.335 | 0.282 | 0.248 |
| | DiffTSR | 22.57 | 0.773 | 0.281 | 0.184 | 45.78 | 16.99 | 0.671 | 0.443 | 0.312 | 0.312 |
| | DiffBIR | 23.76 | 0.731 | 0.245 | 0.196 | 35.99 | 20.82 | 0.770 | 0.275 | 0.234 | 0.293 |
| | OSEDiff | 25.35 | 0.827 | 0.192 | 0.166 | 29.06 | 21.62 | 0.809 | 0.263 | 0.263 | 0.307 |
| | DreamClear | 24.20 | 0.778 | 0.209 | 0.177 | 32.08 | 20.66 | 0.779 | 0.283 | 0.251 | 0.247 |
| | TSD-SR | 22.93 | 0.757 | 0.200 | 0.199 | 37.61 | 19.64 | 0.760 | 0.282 | 0.246 | 0.281 |
| | DiT4SR | 23.45 | 0.774 | 0.204 | 0.158 | 31.73 | 20.12 | 0.779 | 0.250 | 0.208 | 0.266 |
| | TADiSR | 24.68 | 0.795 | 0.362 | 0.227 | 52.13 | 21.57 | 0.798 | 0.366 | 0.347 | 0.375 |
| | Ours | **25.63** | **0.834** | **0.185** | **0.150** | **28.82** | 22.14 | **0.823** | **0.204** | **0.194** | **0.456** |
| | Real-ESRGAN | 22.09 | 0.718 | 0.423 | 0.284 | 89.22 | 18.87 | 0.721 | 0.503 | 0.427 | 0.116 |
| | HAT | 22.66 | 0.738 | 0.451 | 0.299 | 89.58 | 19.27 | 0.738 | 0.533 | 0.484 | 0.123 |
| | MACRONet | 21.00 | 0.716 | 0.422 | 0.254 | 79.67 | 17.01 | 0.680 | 0.553 | 0.382 | 0.123 |
| | SeeSR | 21.79 | 0.731 | 0.310 | 0.228 | 64.31 | 18.87 | 0.717 | 0.396 | 0.308 | 0.157 |
| 14mm (×14.29) | SupIR | 21.38 | 0.681 | 0.447 | 0.268 | 70.09 | 17.99 | 0.690 | 0.487 | 0.383 | 0.110 |
| | DiffTSR | 20.88 | 0.713 | 0.412 | 0.233 | 74.07 | 15.87 | 0.630 | 0.538 | 0.348 | 0.120 |
| | DiffBIR | 21.61 | 0.630 | 0.392 | 0.255 | 69.64 | 18.61 | 0.685 | 0.438 | 0.328 | 0.114 |
| | OSEDiff | 23.11 | 0.768 | 0.274 | 0.214 | 49.76 | 19.44 | 0.740 | 0.385 | 0.334 | 0.127 |
| | DreamClear | 22.45 | 0.719 | 0.364 | 0.260 | 62.95 | 19.14 | 0.725 | 0.440 | 0.377 | 0.124 |
| | TSD-SR | 21.14 | 0.705 | **0.273** | 0.225 | 57.07 | 17.67 | 0.686 | 0.393 | 0.293 | 0.119 |
| | DiT4SR | 20.88 | 0.707 | 0.282 | **0.181** | **47.45** | 17.36 | 0.687 | **0.367** | **0.253** | 0.117 |
| | TADiSR | 22.43 | 0.721 | 0.555 | 0.320 | 98.44 | 19.20 | 0.730 | 0.585 | 0.518 | 0.152 |
| | Ours | **23.41** | **0.774** | 0.301 | 0.209 | 54.74 | **19.73** | **0.748** | 0.374 | 0.322 | **0.160** |
| | RealEsrGAN | 23.99 | 0.790 | 0.248 | 0.194 | 30.60 | 21.25 | 0.786 | 0.302 | 0.266 | 0.371 |
| | HAT | 25.17 | 0.815 | 0.249 | 0.198 | 30.12 | 22.18 | 0.813 | 0.291 | 0.276 | 0.379 |
| | MARCONet | 22.13 | 0.768 | 0.306 | 0.205 | 34.28 | 17.48 | 0.690 | 0.529 | 0.366 | 0.334 |
| | SeeSR | 23.64 | 0.788 | 0.219 | 0.184 | 26.73 | 20.45 | 0.767 | 0.297 | 0.241 | 0.191 |
| | SupIR | 23.62 | 0.754 | 0.308 | 0.207 | 27.91 | 20.10 | 0.756 | 0.344 | 0.287 | 0.236 |
| Total | DiffTSR | 22.41 | 0.767 | 0.300 | 0.189 | 31.25 | 17.00 | 0.665 | 0.452 | 0.313 | 0.317 |
| | DiffBIR | 23.67 | 0.724 | 0.262 | 0.197 | 23.10 | 20.70 | 0.757 | 0.302 | 0.245 | 0.266 |
| | OSEDiff | 25.07 | 0.819 | 0.201 | 0.169 | 20.53 | 21.43 | 0.798 | 0.276 | 0.263 | 0.300 |
| | DreamClear | 24.10 | 0.773 | 0.238 | 0.191 | 21.75 | 20.50 | 0.768 | 0.317 | 0.276 | 0.214 |
| | TSD-SR | 22.79 | 0.757 | 0.207 | 0.194 | 24.08 | 19.42 | 0.748 | 0.296 | 0.249 | 0.266 |
| | DiT4SR | 23.16 | 0.767 | 0.215 | 0.159 | 20.58 | 19.73 | 0.760 | 0.273 | 0.216 | 0.237 |
| | TADiSR | 24.61 | 0.796 | 0.203 | 0.160 | 36.61 | 21.59 | 0.799 | 0.360 | 0.336 | 0.384 |
| | Ours | **25.48** | **0.830** | **0.196** | **0.156** | **20.01** | **22.22** | **0.814** | **0.228** | **0.212** | **0.430** |

Table 7: Statistics of dataset size and line count in subsets of UltraZoom-ST.

| Subset | image count | line count | mean lines/img | img > 5 lines |
|--------|-------------|------------|----------------|---------------|
| 14 mm | 1,439 | 15,073 | 10.47 | 754 |
| 35 mm | 1,798 | 17,263 | 9.60 | 867 |
| 85 mm | 1,799 | 17,339 | 9.64 | 869 |

Table 8: Efficiency analysis.

| Methods | Flops (GFLOPs) | Speed (ms) | OCR-A |
|---------|----------------|------------|-------|
| HAT | 6670.32 | 1086.68 | 0.379 |
| DiffTSR | 58502.14 | 8610.59 | 0.317 |
| DiT4SR w/o llava | 160787.14 | 17385.14 | 0.237 |
| DreamClear w/o llava | 412843.67 | 83193.81 | 0.214 |
| TADiSR | 4497.96 | 342.32 | 0.384 |
| TIGER (ours) Stage 1 | 6215.85 | 663.5 | 0.430 |
| TIGER (ours) Stage 2 | 3734.01 | 360.06 | |

## A.2 DATASET COLLECTION

We use VIVO X200 Ultra to collect images for 4 separate focal lengths (14 mm, 35 mm, 84 mm, and 200 mm). We first use PP-OCRV5 Cui et al. (2025) for the rough annotation, then we manually filter the images and annotations. During the filtering and annotation, each image undergoes the following rules:

- Width or height of the image should be no less than 256.

- Height of the text should not be less than 32 pixels.

- Score of OCR recognition of the text should not be lower than 0.9.

- Content of the text should not be empty or consist solely of whitespace.

## A.3 MORE IMPLEMENTATION DETAILS

**Dataset Settings.** For training, we combine a synthetic dataset with real paired datasets (Real-CE (Ma et al., 2023) and UltraZoom-ST). Our synthetic dataset builds upon LSDIR (Li et al., 2023b), containing 27,000 triplets $(x_H, x_L, x_m)$. We render text on the GT of LSDIR and the corresponding text mask using the LBTS (Tang et al., 2023), then apply Real-ESRGAN degradation Wang et al. (2021) to generate LR images. Since there are misaligned image pairs in Real-CE (Hu et al., 2025; Zhang et al., 2024a) and text lines not annotated, we filter out misaligned image pairs and reannotate the images manually. In the end, we obtain 337 training pairs and 188 testing pairs from the Real-CE dataset. Images under 13mm and 52mm focal lengths are considered as LR $x_L$ and GT $x_H$, respectively. Following Hu et al. (2025), we use SAM-TS (Ye et al., 2024) to obtain the text mask $x_m$ from $x_H$. UltraZoom-ST triplets $(x_H, x_L, x_m)$ are processed identically. For stage 1 training, we use cropped text regions from the dataset, while for stage 2, we use full images. Following Tuo et al. (2024), we use PP-OCRv3 Li et al. (2022) to extract strings from the text region of predicted SR for evaluation of OCR-A.

**Training Details.** For stage 1, we build our model based on the IDM baseline of DiffTSR (Zhang et al., 2024a). We set $\lambda_{td}$, $\lambda_{Seg}$, $\lambda_{Focal}$, and $\lambda_{Dice}$ as 1, 0.1, 20, and 1 respectively. In stage 2, our model is built upon Stable Diffusion 3.5 medium (Esser et al., 2024) and follows a similar tile-based inference strategy to TADiSR (Hu et al., 2025) and SUPIR (Yu et al., 2024). The timestep $t$ is set to 150. $\lambda_{l2}$, $\lambda_{LPIPS}$, and $\lambda_{edge}$ are set to 1, 5, 100 when using the synthetic dataset for pretraining. Then, $\lambda_{edge}$ is set to 0 and the remaining coefficients are kept unchanged when training on real paired datasets. We train both stages of the model using the AdamW (Loshchilov & Hutter, 2017) optimizer and set the learning rate to $5 \times 10^{-5}$ and $5 \times 10^{-6}$ for stages 1 and 2 separately. All experiments are conducted on NVIDIA H20 GPUs. For stage 1, we train the model with synthetic and real datasets for 8 epochs, and then use synthetic data to finetune the model for 2 epochs. For

stage 2, we first pretrain the model on the synthetic dataset for 50 epochs. Then we train on the real paired datasets for 50 epochs.

### A.4 ADDITIONAL RESULTS

We provide detailed quantitative results of UltraZoom-ST in Table 6. Additional qualitative results are provided in Fig. 6.

### A.5 STATIC AND EXAMPLES OF ULTRAZOOM-ST

In Table 7, we provide detailed statistics on the composition of the UltraZoom-ST dataset. Additionally, in Fig. 7, we present some example images from the dataset.

### A.6 EFFICIENCY ANALYSIS

As shown in Table 8, stage 1 is a standard diffusion process that takes multiple steps in inference, the efficiency of our model may be suboptimal compared to one-step methods. However, our method achieves state-of-the-art performance in OCR-A, which cannot be easily obtained by extending inference time.

### A.7 DISCUSSIONS AND LIMITATIONS

Our method leverages an OCR model to localize and interpret text in low-resolution images. In cases of severe degradation, however, OCR fails to recognize the text, which leads to failures in our method as well. Human-provided annotations could potentially mitigate this limitation.

A further challenge arises in real-world deployment. Because our text restoration model is diffusion-based, it requires multiple inference steps, which hinders real-time performance. Although our model reconstructs text structures more plausibly than prior approaches, it does not yet satisfy the speed demands of practical applications. Strategies such as model distillation and quantization offer promising directions to address this bottleneck.

This work is presented as an academic study of scene text image super-resolution in real-world conditions. Still, given its relevance to many applications, it carries both potential benefits (e.g., enhancing image quality in consumer devices) and risks (e.g., exposing private information from photos on social media). On balance, the societal benefits far outweigh the risks, particularly since complementary techniques such as inpainting and watermarking can be used to safeguard sensitive information.

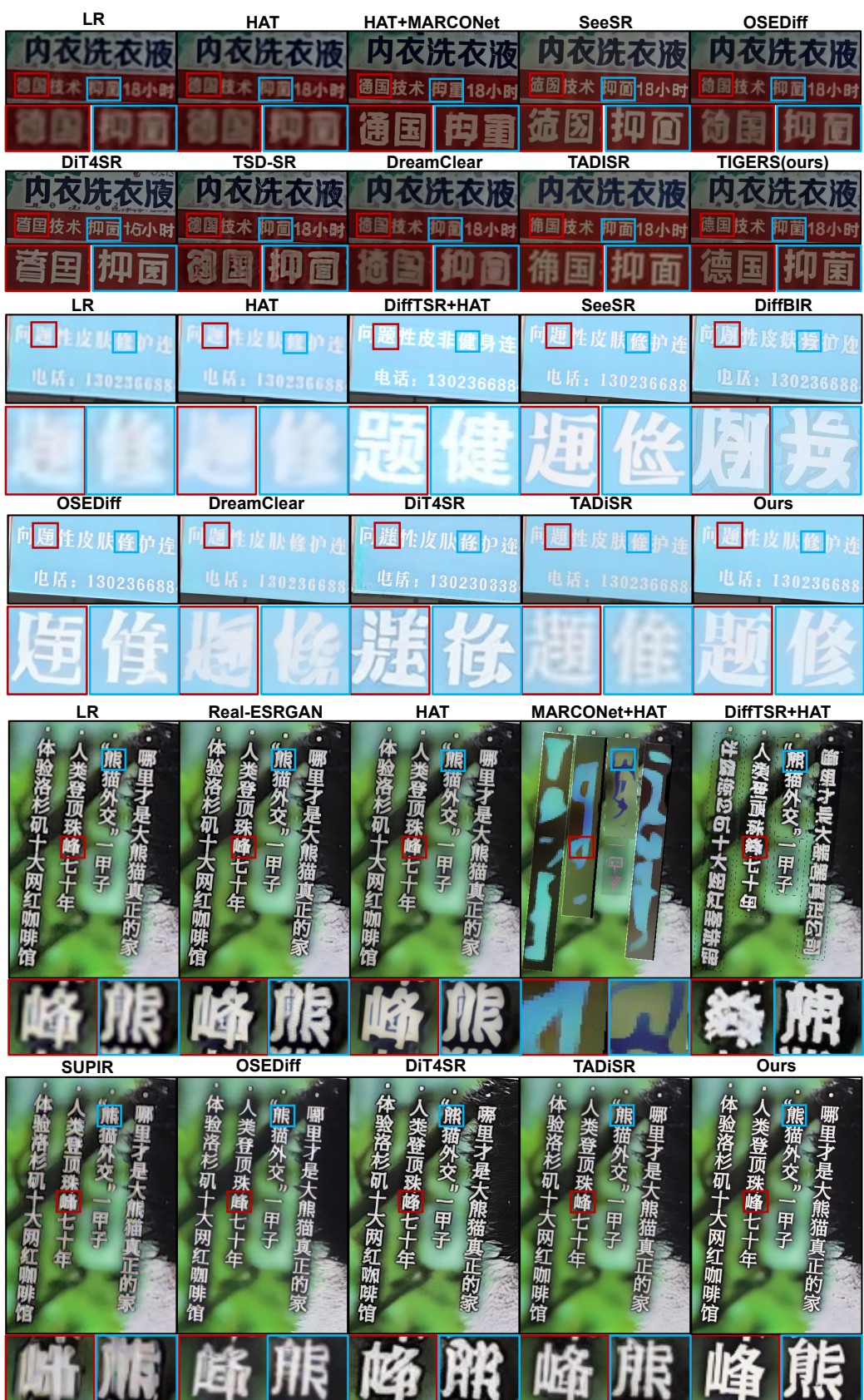

Figure 6: Additional qualitative results of TIGER.

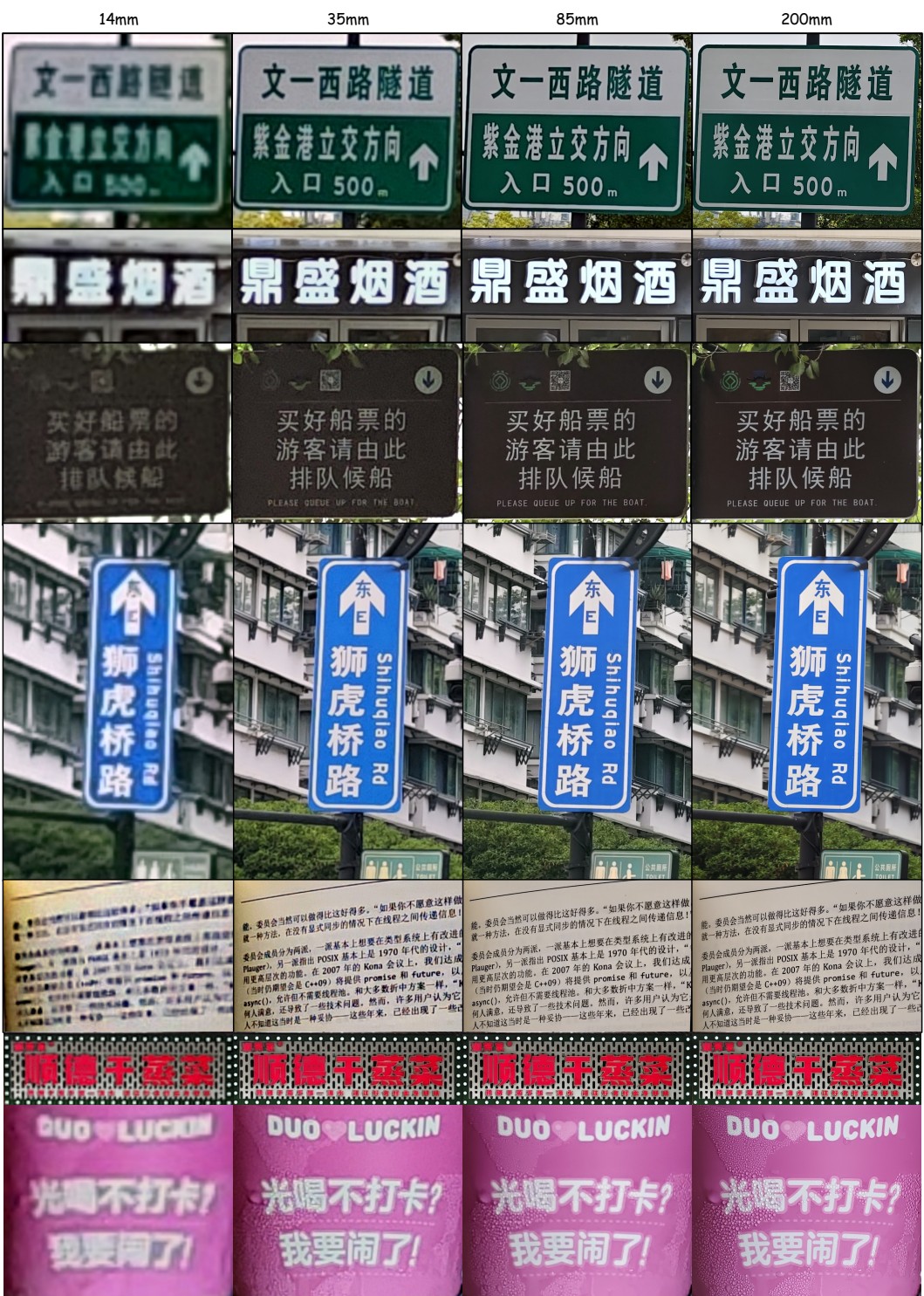

Figure 7: Detailed examples of UltraZoom-ST.

