# OpenReview forum: "Restore Text First, Enhance Image Later: Two-Stage Scene Text Image Super-Resolution with Glyph Structure Guidance"
_ICLR.cc/2026/Conference — ICLR 2026 Conference Withdrawn Submission_

### Official Review · Reviewer_iuBT · 2025-10-28

**Soundness:** 3
**Presentation:** 3
**Contribution:** 2
**Rating:** 4
**Confidence:** 4

**Summary:**

The paper introduces TIGER (“Text–Image Guided supEr-Resolution”), a two-stage scene text SR framework following a “restore text first, enhance image later” paradigm. Stage-1 reconstructs glyph structures (a text mask) with an OCR-conditioned diffusion pipeline; Stage-2 performs full-image SR guided by that mask via a ControlNet-like conditioning, with additional edge losses to preserve strokes. The authors also contribute UltraZoom-ST, a real-world dataset with extreme zoom (×14.29) and multi-focal alignment, and report SOTA results on Real-CE and UltraZoom-ST across image quality (PSNR/SSIM/LPIPS/DISTS/FID) and a text accuracy metric (OCR-A, Levenshtein ratio).

**Strengths:**

Clear decoupling and strong motivation.
Separating glyph restoration (with semantic conditioning from OCR) from later full-image enhancement directly targets the field’s chronic trade-off between readability and perceptual quality and is well motivated for complex scripts (e.g., Chinese). The “text mask → ControlNet guidance” design is coherent.

Technically sound Stage-1 and Stage-2 pipelines.
Stage-1 uses VAE latents, a UNet denoiser, and dual branches (appearance and structure) with a segmentation-oriented loss (MSE+Focal+Dice); Stage-2 adds glyph-aware conditioning and edge loss to preserve strokes—reasonable choices for text fidelity.

Dataset contribution and alignment pipeline.
UltraZoom-ST introduces stronger real degradations than Real-CE and a coarse-to-fine alignment across focal lengths; statistics and collection rules are documented, enabling more realistic evaluations.

Comprehensive metrics with text-region breakdown.
Beyond full-image metrics, authors report text-region (cr) metrics and OCR-A using Levenshtein, which better capture the task goal (readability).

Solid empirical gains and ablations.
Tables 2–3 show improvements over diverse baselines; the ablation of text mask sources (Std-Font, SAM-TS, LDM vs. TIGER) shows that TIGER’s Stage-1 gives the best guidance to Stage-2.

**Weaknesses:**

Dependence on OCR correctness with limited analysis.
Stage-1 assumes reliable text localization and transcription to condition glyph restoration, but OCR degrades under severe LR noise; the paper acknowledges failures when OCR fails, yet offers no quantitative sensitivity to OCR errors or calibration analysis (e.g., how often wrong OCR pushes Stage-1 toward wrong glyphs). A robustness study to OCR precision/recall or confidence thresholds is needed.

C2. Dataset realism and alignment validation.
UltraZoom-ST is valuable, but alignment under ×14.29 zoom is non-trivial; the coarse-to-fine pipeline description is clear, yet there is no quantitative alignment accuracy (e.g., keypoint residuals, PSNR between warped LR and HR) or manual checks across the test split. Given SR metrics are pixel-based, even small misalignments can inflate/deflate results. Please provide alignment error distributions and examples of failure modes.

C3. Baseline fairness and representativeness.
Many baselines are fine-tuned on the new training sets (“ensure fairness”), but precise tuning budgets, prompts/settings for diffusion/DiT variants, tiling policies, and early-stop criteria are not fully specified in the main text. Report per-method hyperparameter grids and compute parity. For MARCONet/DiffTSR integrations into HAT, clarify integration details to avoid inadvertent handicaps.

C4. Statistical rigor and significance.
The tables report single-number results without mean±std over multiple runs or significance tests. Given differences can be modest in some settings, ICLR standards typically expect variance reporting on key metrics (especially OCR-A). Provide CIs or paired tests on common subsets.

C5. Efficiency framing vs. two-stage cost.
Table 8 shows large FLOPs and latency for diffusion-based systems; while authors argue the gains in OCR-A justify costs, there is no tokenized inference-time breakdown (Stage-1 vs. Stage-2 vs. OCR pre-/post-processing). A deployment-oriented analysis (e.g., one-step distillation, mask caching, or when to skip Stage-1) would make the case stronger.

C6. Generalization beyond Latin/Chinese & domain transfer.
Claims about multilingual robustness are implicit; UltraZoom-ST contains Chinese and multi-line scenarios, but the paper does not report cross-script or cross-domain transfer (e.g., signage vs. documents vs. LED displays). Report per-language or per-script performance breakdowns, if available.

**Questions:**

Questions by section
Method (Stage-1 Text Restoration)
1) Text-mask branch supervision. You propose a segmentation-oriented loss (MSE+Focal+Dice) using an estimate x'_m decoded from denoised z_0^m. Please provide calibration plots (e.g., Dice vs. confidence) and visualizations of mask noise across difficulty levels; also, report ablation of λFocal, λDice.
2) OCR conditioning. Which OCR outputs are used (string only vs. char boxes)? How do you handle OCR uncertainty (e.g., n-best)? Please quantify performance drop when OCR recall is artificially reduced by {10, 20, 30}% on the validation set.

Method (Stage-2 Image Enhancement)
1) ControlNet conditioning strength. Show a sweep of the conditioning scale from mask (how strongly to inject the structure); report failure cases where over-strong guidance causes haloing or edge ringing.
2) Edge loss (Sobel). Justify Sobel over canny/Laplacian; include a replacement ablation and a per-size study for small glyphs (<16 px height).

Dataset & Alignment
1) Alignment QA. Provide quantitative homography residuals or SSIM after warping per focal pair; attach a table of alignment outlier rates (RANSAC inliers %) to support pixel-wise metric validity.
2) Collection rules. The appendix lists OCR score ≥0.9 during filtering; this may bias the dataset toward easier readable cases. Please quantify the distribution shift introduced by this filter (e.g., compare to raw captures).

Metrics & Evaluation
1) OCR-A reliability. OCR-A uses Levenshtein ratio; specify the OCR engine version and settings used at eval time, and report inter-OCR agreement (PP-OCRv3 vs. another recognizer) to ensure metric robustness.
2) Per-level reporting. You mention per-zoom results in the appendix; surface confidence intervals per zoom level (×2.35 / ×5.71 / ×14.29) and per text-height bin to reveal where TIGER helps most.

Ablations & Variants
1) Mask source study is good (Std-Font / SAM-TS / LDM / TIGER). Please add a no-mask and a soft-mask variant (probabilistic maps) to test whether soft structural priors outperform binary masks in Stage-2.
2) Two-phase training. Provide curves showing where Phase-2 (synthetic-only finetune) improves mask quality but does not overfit synthetic textures; include FID/LPIPS on real before/after Phase-2 for Stage-1 outputs.

---

### Official Review · Reviewer_6ReV · 2025-10-31

**Soundness:** 3
**Presentation:** 3
**Contribution:** 2
**Rating:** 4
**Confidence:** 3

**Summary:**

The paper proposes TIGER, a two-stage “text-first, image-later” framework for scene text super-resolution. It first restores glyph structures using a diffusion-based refiner and then enhances the full image guided by text masks. A new dataset, UltraZoom-ST, is collected using multi-focal real captures to simulate realistic degradations.

**Strengths:**

1. The “text-first, image-later” idea is interesting and provides a clear conceptual separation between text restoration and image enhancement.

2. The authors successfully trained a two-stage model and presented several good qualitative visualizations.

3. The paper introduces a new approach for constructing real paired data and contributes a dataset, UltraZoom-ST, to the community.

**Weaknesses:**

1. The system heavily depends on OCR for text detection, and any detection errors can directly degrade the overall performance.

2. The two-stage training process is complex and increases implementation difficulty.

3. The methodological novelty is limited, as the proposed two-stage text-first paradigm mainly combines existing diffusion-based restoration and OCR-guided enhancement strategies rather than introducing a fundamentally new framework.

4. The quantitative improvements are small, with OCR-A increasing by only 0.026 on Real-CE and 0.046 on UltraZoom-ST, showing limited absolute gains despite higher complexity.

5. Although coarse-to-fine geometric registration is used, multi-focal real captures still cannot guarantee perfect pixel alignment between LR and HR pairs.

6. The dataset relies on images captured by the VIVO X200 Ultra smartphone, whose built-in ISP pipeline introduces lens distortion, denoising, and sharpening, leading to potential bias from device-specific enhancement artifacts.

**Questions:**

1. Since the system heavily depends on OCR for text detection, how robust is the model when OCR fails or misdetects small or occluded text?

2. Given that prior works such as TADiSR and DiffTSR already integrate text understanding within the image restoration process, how does the proposed sequential text-first design represent a substantive methodological innovation rather than a rearrangement of existing components?

3. With the quantitative improvements being relatively small, how significant are these gains in real-world perception or downstream OCR tasks?

4. Since the dataset relies on images captured by the VIVO X200 Ultra with in-camera ISP processing, how do the authors control or mitigate the bias introduced by device-specific enhancement and denoising algorithms?

---

### Official Review · Reviewer_hBem · 2025-11-01

**Soundness:** 3
**Presentation:** 2
**Contribution:** 2
**Rating:** 6
**Confidence:** 4

**Summary:**

This paper proposes TIGER, a two-stage framework for scene text image super-resolution, featuring a “text-first, image-later” paradigm to decouple glyph restoration from image enhancement.
Specifically, stage 1 (Text Restoration Stage) extract the text regions from the LR input and reconstructs glyph structures via a denoising U-net; stage 2 enhances global image quality conditioned on the restored text masks by adopting a glyph-aware ControlNet network.

The authors also introduce UltraZoom-ST, a new dataset with real-world text images captured under four fixed focal lengths (14 mm, 35 mm, 85 mm, 200 mm), enabling extreme zoom (×14.29). Experiments on Real-CE and UltraZoom-ST demonstrate consistent gains achieved by TIGER in both image quality (PSNR/SSIM/LPIPS/DISTS/FID) and text readability (OCR-A).

**Strengths:**

1. The paper introduces a novel two-stage framework, decoupling glyph restoration from image enhancement to solve the trade-off between text readability and image fidelity in STISR. The “text-first, image-later” paradigm is intuitive and practically useful.
2. Experiments showed that TIGER achieves consistent SOTA results across multiple benchmarks and metrics, especially in OCR-A accuracy and FID, with clear quantitative and qualitative evidence.
3. The proposed UltraZoom-ST dataset provides a large amount of images containing non-Latin letters, which fills an existing gap in STISR benchmarks.

**Weaknesses:**

1. While the two-stage framework seems to be intuitive and useful, the architecture itself resembles TADiSR a lot. It seems that TIGER splits the diffusion process in TADiSR into two stages, where text mask and SR image are restored in order instead of jointly. This improves image fidelity and text readability, however, the computational cost could be 2x higher.

2. Most baselines in experiments are not SR methods specialized for scene text image, except DiffTSR and TADiSR. In addition, methods like DiffTSR are not built for restoring a full scene text image, but for cropped image that only contains a single textline. The comparison could be unfair.

3. While the proposed UltraZoom-ST dataset contains LR images shot in different focal lengths, only LR 3 (14mm) images are challenging enough from the perspective of human view. As shown in Table 6, even methods not spacialized for STISR can achieve outstanding performance on LR 1 (85mm) images.

4. Lack of degradation like distortion, motion blur in the proposed UltraZoom-ST dataset, which are commonly shown in former STISR dataset like TextZoom, and would significantly make restoration harder.

**Questions:**

1. How was DiffTSR applied to this task? Were the full scene text images directly fed to DiffTSR, or fed after cropping?
2. Sec A.2 said that the score of OCR recognition of the text in original HR image should not be lower than 0.9. What about the initial OCR accuracy in LR images? Table 2,3,6 didn't show them, only showing accuracy of baselines.

---

### Official Review · Reviewer_13Rr · 2025-11-01

**Soundness:** 2
**Presentation:** 2
**Contribution:** 2
**Rating:** 2
**Confidence:** 4

**Summary:**

This paper introduces TIGER (Text-Image Guided super-Resolution), a progressive two-stage paradigm for scene text super-resolution that addresses the persistent trade-off between maintaining readability and ensuring high image quality. The key insight is that text and non-text regions should be handled differently, with text structures reconstructed first before guiding full-image restoration.

**Strengths:**

a. Extensive experiments on both Real-CE and UltraZoom-ST demonstrate the superior super-resolution performance of the proposed model, particularly in preserving text structure fidelity.

​b. This paper exploits variations in camera focal length to construct 5,036 real-world training pairs, forming the UltraZoom-ST dataset. The dataset features multi-line text instances, diverse scenarios, and varied lighting conditions.

**Weaknesses:**

a. Incomplete citation: The innovative contributions of this paper highly overlap with [1], as both studies employ a foreground text prior to guide global text image inpainting. However, the presented work fails to cite or discuss the relevant work, thereby undermining the perceived novelty of the proposed method.

​	b. Critical dependency on pre-trained OCR models: The 'Text Restoration process' in the stage 1 heavily relies on the performance of the OCR model. When the OCR model fails to function properly on heavily degraded text images, the proposed TIGER framework may become inoperative.

​	c. The reconstruction quality of VAE: The two-stage restoration process in TIGER heavily relies on the capabilities of the VAE. However, during the VAE reconstruction phase, small-sized Chinese characters frequently experience reconstruction failures, particularly when utilizing the SD3.5 pre-trained VAE employed in this study. The authors did not adequately address this limitation in their analysis.

​	d. Theoretical validity of the MMDIT architecture: This paper adopts MMDIT as its backbone in the stage 2. However, MMDIT was originally designed to address bimodal fusion problems involving both visual and textual modalities. Since the stage 2 in this paper operates as a unimodal image task, the architectural choice of MMDIT lacks a solid theoretical foundation for this specific application.

​	e. Computational Efficiency Concerns: The TIGER framework relies on collaborative operations across multiple sub-models. Specifically, the SD3.5 backbone in stage 2 alone contains 2.5 billion parameters and requires multi-step inference procedures. This architectural design leads to suboptimal computational efficiency, yet the authors do not provide comprehensive analysis regarding this computational burden in their methodology.

​	Reference:

​	[1] Zhu, et al. Text Image Inpainting via Global Structure-Guided Diffusion Models. AAAI, 2024.

**Questions:**

​	a. One critical objective of scene text image super-resolution (STISR) is to enhance the recognition accuracy of downstream OCR systems. However, the architecture of TIGER inherently requires accurate OCR predictions as a prerequisite for executing STISR tasks. In other words, the performance ceiling of this model is fundamentally constrained by the capabilities of the pre-trained OCR system it relies on. This dependency, in my view, represents the most questionable design choice. The author needs to provide relevant detailed discussions.

​	b. Based on the "weaknesses" section, provide an analysis of the model structure selection and computational efficiency.

---

### Note · Authors · 2025-11-14

I have read and agree with the venue's withdrawal policy on behalf of myself and my co-authors.